# SPR$^2$Q: Static Priority-based Rectifier Routing Quantization for Image Super-Resolution

**Jingwei Xin**[1], **Wenhao Li**[1], **Nannan Wang**[1]*, **Jie Li**[2], **Xinbo Gao**[2]

[1]School of Communication Engineering, Xidian University
[2]School of Electronic Engineering, Xidian University
{jwxin, nnwang}@xidian.edu.cn
wenhali@stu.xidian.edu.cn
{leejie, xbgao}@mail.xidian.edu.cn

## Abstract

Low-bit quantization has achieved significant progress in image super-resolution. However, existing quantization methods show evident limitations in handling the heterogeneity of different components. Particularly under extreme low-bit compression, the issue of information loss becomes especially pronounced. In this work, we present a novel low-bit post-training quantization method, namely static priority-based rectifier routing quantization (SPR$^2$Q). The starting point of this work is to attempt to inject rich and comprehensive compensation information into the model before the quantization , thereby enhancing the model's inference performance after quantization. Firstly, we constructed a low-rank rectifier group and embedded it into the model's fine-tuning process. By integrating weight increments learned from each rectifier, the model enhances the backbone network while minimizing information loss during the lightweighting process. Furthermore, we introduce a static rectifier priority routing mechanism that evaluates the offline capability of each rectifier and generates a fixed routing table. During quantization, it updates weights based on each rectifier's priority, enhancing the model's capacity and representational power without introducing additional overhead during inference. Extensive experiments demonstrate that the proposed SPR$^2$Q significantly outperforms the state-of-the-arts in five benchmark datasets, achieving PSNR improvements of 0.55 and 1.31 dB on the Set5($\times$2) dataset under 4-bit and 2-bit settings, respectively.

## 1 Introduction

With the rapid development of deep learning, image super-resolution (SR) models have achieved remarkable breakthroughs in performance (Dong et al., 2015; Guo et al., 2025). However, their high computational and storage costs severely limit deployment on real-world devices. Consequently, how to achieve efficient inference while maintaining accuracy has become a critical research focus, among which low-bit quantization stands out as a highly promising solution (Han et al., 2016; Courbariaux et al., 2016; Gholami et al., 2021). Low-bit quantization compresses floating-point parameters of neural networks into lower-bit representations, thereby reducing model size and latency while preserving accuracy and enabling hardware acceleration.

Quantization methods can generally be divided into quantization-aware training (QAT) and post-training quantization (PTQ) (Choi et al., 2017; Jacob et al., 2018b). Although QAT is widely recognized for minimizing accuracy loss (Mishra & Marr, 2018), it often requires high training costs and long training time—sometimes even heavier than training the original full-precision model. In contrast, PTQ completes quantization after training by adjusting quantizer parameters or calibrating weights/activations (Nagel et al., 2019; Banner et al., 2019), without retraining the model. Thus, PTQ offers low training cost and fast deployment, but it tends to suffer from significant accuracy degradation under ultra-low-bit settings (Nahshan et al., 2021; Li et al., 2021).

---

*Corresponding author.

Despite the progress of post-training quantization (PTQ) methods across diverse architectures such as Transformers and Mamba (Gong et al., 2025), existing solutions exhibit significant shortcomings in their adaptability across different architectures and domains. This is primarily manifested in two aspects: First, while current low-bit quantization methods have been successfully applied to Transformer-based super-resolution (SR) models like SwinIR (Liang et al., 2021; Liu et al., 2024), they fail to adapt to the unique computational paradigm of the Mamba architecture (Gu & Dao, 2023). Specifically, these methods struggle to address the error accumulation and numerical sensitivity issues arising from Mamba's recurrent state and dynamic gating mechanisms. This ultimately leads to a substantial degradation in the ability of the quantized model to restore fine image details. Second, most existing Mamba quantization methods have been validated primarily on tasks such as classification or language modeling (Xu et al., 2025; Cho et al., 2025). Super-resolution, however, is exceptionally sensitive to pixel-level precision and the fidelity of local textures. Consequently, porting these methods to the SR domain often yields unsatisfactory results, as illustrated in Figure 2, where they fail to meet the stringent fidelity requirements and cause blurred details and texture loss.

These limitations indicate that PTQ, which merely optimizes quantizer parameters, is insufficient to overcome the challenges posed by aggressive low-bit compression. We argue that achieving extreme low-bit performance requires not only better quantizers but, more importantly, enabling the model itself to actively adapt to the quantization process through a small set of trainable parameters. By injecting complementary information into the model before quantization, the substantial information loss introduced by aggressive compression can be effectively mitigated.

To this end, our SPR$^2$Q framework achieves this active model adaptation on two fronts. First, inspired by the idea of LoRA (Hu et al., 2022), we fuse the weight increments from low-rank rectifier modules into the backbone network pre-quantization. This design ensures that the supplementary information, learned to compensate for quantization error, is incorporated into the quantization process as prior knowledge, fundamentally mitigating information loss while preserving the full inference acceleration benefits. Furthermore, to enhance the diversity of compensation information, SPR$^2$Q introduces a rectifier priority routing strategy. In this design, multiple rectifier modules are trained as a rich group of information compensators. A static routing table is then constructed through offline evaluation, assigning priorities to each rectifier. During inference, the model updates its weights according to the rectifier priorities, thereby expanding its representational space and achieving significant performance improvements without incurring any additional computational cost. Our contributions can be summarized as follows:

- We introduce SPR$^2$Q, a novel quantization method addressing low-bit quantization challenges in super-resolution. Its architecture is composed of two synergistic components: a pre-quantization fusion rectifier module for injecting learnable compensation, and a static rectifier priority routing that injects pre-evaluated compensation into the model.

- SPR$^2$Q's methodology begins with pre-quantization fusion, embedding rectifier-learned compensation into the backbone to mitigate information loss. Subsequently, a Rectifier group is constructed, and the static rectifier priority routing mechanism updates weights by rectifier priority, providing the model with diverse information for complex detail recovery.

- Extensive experiments validate our state-of-the-art performance on challenging low-bit super-resolution tasks. On the MambaIRv2 model, SPR$^2$Q significantly outperforms multiple leading techniques, achieving PSNR improvements of up to 0.55 and 1.31 dB on Set5 ($\times2$), under 4-bit and 2-bit quantization.

## 2 RELATED WORK

### 2.1 IMAGE SUPER-RESOLUTION

Deep learning has significantly advanced the field of image super-resolution (SR). Early approaches were dominated by convolutional neural networks (CNNs), ranging from the pioneering SRCNN (Dong et al., 2015) to EDSR (Lim et al., 2017), which improved performance by introducing residual connections and increasing model capacity. Subsequent works further explored the potential of CNNs with more sophisticated architectures. For instance, RDN (Zhang et al., 2018b) leverages residual dense blocks to fully exploit hierarchical features, while RCAN (Zhang et al., 2018a) introduces channel attention mechanisms to learn more discriminative features, both significantly en-

hancing reconstruction quality. As research progressed, the limitations of CNNs in capturing long-range dependencies and global context became evident. To address this, Transformers (Vaswani et al., 2017) were introduced into the SR domain. Early attempts, IPT (Chen et al., 2021), demonstrated the great potential of pure Transformer architectures for image processing tasks, though their high computational cost limited practical applicability. Later works, including SwinIR (Liang et al., 2021) and ATD (Zhang et al., 2024), incorporated efficient designs such as window attention to model long-range dependencies while substantially reducing computational overhead, achieving state-of-the-art performance across multiple benchmarks. The success of Transformer-based SR models highlights the advantages of self-attention mechanisms in capturing spatial correlations over large receptive fields. More recently, the emergence of the Mamba (Gu & Dao, 2023) architecture has driven SR research toward state space model (SSM)-based frameworks. Representative works, including MambaIR (Guo et al., 2024) and MambaIRv2 (Guo et al., 2025), exploit the efficient sequence modeling capabilities of state space models to capture long-range dependencies, achieving high-quality reconstruction with reduced computational overhead. These Mamba-architecture models achieve superior reconstruction quality compared to Transformer-based methods while significantly reducing computational overhead, showcasing Mamba's unique advantage in balancing efficiency and performance.

## 2.2 MODEL QUANTIZATION

Quantization methods can be broadly categorized into quantization-aware training (QAT) and post-training quantization (PTQ). QAT is capable of minimizing performance degradation and was thus widely adopted in early studies. Representative works such as PAMS (Hirano et al., 2023) and CADyQ (Hong et al., 2022) primarily focused on lightweight compression for CNNs, aiming to reduce computational and storage overhead while preserving reconstruction quality. However, these approaches typically incur substantial training costs, often requiring as much or even more training time than the original models. To address this challenge, PTQ methods were introduced, which directly operate on pretrained models and only require boundary calibration of quantizers. DBDC+Pac (Tu et al., 2023) is the first PTQ method specifically designed for image super-resolution, achieving superior performance on EDSR (Lim et al., 2017) and SRResNet (Ledig et al., 2017), thereby demonstrating the potential of PTQ for SR tasks. With the rise of Transformer-based SR models, researchers have also begun exploring PTQ tailored to these architectures. For instance, 2DQuant (Liu et al., 2024) achieves excellent results on SwinIR, showing that carefully designed boundary calibration and quantization strategies can effectively mitigate the accuracy degradation caused by low-bit quantization in Transformers. Nevertheless, for more complex and emerging architectures such as Mamba-based SR models, existing quantization research still mainly focuses on large language models and image classification, leaving quantization for SR largely underexplored.

## 3 METHOD

The core principle of existing Post-Training Quantization (PTQ) methods for Super-Resolution (SR) is to find optimal quantizer parameters for a given set of fixed, pre-trained weights. This process is typically modeled using a Quantization-Dequantization (QDQ) function (Jacob et al., 2018a):

$$\hat{x} = \mathrm{clip}(x, a, b), \quad s = \frac{b - a}{2^n - 1}, \quad x_q = \mathrm{round}\left(\frac{\hat{x} - a}{s}\right) \cdot s + a, \tag{1}$$

where $n$ is the bit width and $a, b$ denote the clipping range. The quantizer clips the input to $[a, b]$, scales and rounds it to discrete levels, and then dequantizes it to floating point. Existing PTQ methods mainly differ in how $a$ and $b$ are determined. Whether using static statistics (Nagel et al., 2019) or optimization-based schemes (Li et al., 2021), they aim to minimize the quantization error $|x - x_q|$ by tuning these parameters. However, such quantizer-only approaches ignore the model's ability to adapt to quantization. To overcome this, we propose SPR$^2$Q, which introduces lightweight learnable rectifiers to compensate quantization errors.

### 3.1 PRE-QUANTIZATION FINE-TUNING WITH FUSED RECTIFIER

To enable proactive model rectification, we introduce a Pre-Quantization Fine-tuning with Fused Rectifier (PQFR) mechanism. The core idea is to augment the original weights $W$ with a

lightweight, trainable rectifier, $\Delta W$, before quantization. This rectifier is parameterized by two low-rank matrices, $A \in \mathbb{R}^{r \times d_{in}}$ and $B \in \mathbb{R}^{d_{out} \times r}$. Fusing the rectifier yields a new, more quantization-robust weight matrix $W'$, which becomes the actual target for quantization. This process is formulated as:

$$W' = W + \Delta W, \quad \Delta W = BA, \tag{2}$$

$$W'_q = Q_{a,b}(W') = Q_{a,b}(W + BA), \tag{3}$$

where $W \in \mathbb{R}^{d_{out} \times d_{in}}$ represents the frozen pre-trained weights. The pseudo-quantization operator $Q_{a,b}(\cdot)$ is defined in Eq. 1, featuring trainable clipping bounds $a$ and $b$.

We jointly optimize the rectifier parameters $(A, B)$ and quantizer parameters $(a, b)$ using a hybrid loss function. This loss integrates a pixel-level reconstruction objective with a fine-grained block-level feature alignment objective, enabling compensation at both global and local levels.

The first component, the pixel-level loss function (Dong et al., 2015), ensures reconstruction fidelity by minimising the difference between the quantised model output and the full-precision model output image:

$$\mathcal{L}_{\text{pixel}} = \mathbb{E}_{(x,y_{\text{FP}}) \sim \mathcal{D}_{\text{train}}} \left[ \| f_q(x) - y_{\text{FP}} \|_1 \right], \tag{4}$$

The second component, the block-wise feature alignment loss, encourages the quantized model to mimic the full-precision (FP) model at the level of individual computational blocks (Hinton et al., 2015). Instead of applying feature distillation only at coarse or stage-level granularity, we impose alignment constraints on each block, ensuring that local discrepancies are compensated progressively across network depth. Formally:

$$\mathcal{L}_{\text{feature}} = \mathbb{E}_{x \sim \mathcal{D}_{\text{train}}} \left[ \sum_{l=1}^{L} \| \phi_l(f_q(x)) - \phi_l(f_{\text{FP}}(x)) \|_2^2 \right], \tag{5}$$

where $\phi_l(\cdot)$ denotes the feature map extracted from the $l$-th block, and $L$ is the total number of distilled blocks. This design not only captures channel-level statistical consistency, but also provides fine-grained alignment at the block level, thereby mitigating distortions introduced by quantization at a more microscopic scale.

The final training objective is a weighted combination of the pixel-level reconstruction loss and the block-wise feature alignment loss:

$$\mathcal{L} = \mathcal{L}_{\text{pixel}} + \lambda \mathcal{L}_{\text{feature}}, \tag{6}$$

This design ensures that the model simultaneously preserves output fidelity while progressively reducing quantization-induced discrepancies across intermediate blocks.

During backpropagation, we adopt the Straight-Through Estimator (STE) (Bengio et al., 2013) to approximate the gradient of the non-differentiable rounding function in $Q_{a,b}(\cdot)$. This allows gradients to flow through the quantizer while optimizing both the rectifiers and the clipping bounds.

The gradients of $\mathcal{L}$ then update both the rectifier parameters and the quantizer parameters in a unified manner. For the low-rank rectifier matrices $(A, B)$, the gradients are computed as:

$$\frac{\partial \mathcal{L}}{\partial A} = B^\top \frac{\partial \mathcal{L}}{\partial W'}, \quad \frac{\partial \mathcal{L}}{\partial B} = \frac{\partial \mathcal{L}}{\partial W'} A^\top, \tag{7}$$

These updates allow the rectifier to directly absorb error signals and provide effective compensation for the perturbed quantized weights.

At the same time, the trainable clipping bounds $(a, b)$ are also refined through gradient-based updates:

$$\frac{\partial \mathcal{L}}{\partial v} = \frac{\partial \mathcal{L}}{\partial W'_q} \cdot \frac{\partial W'_q}{\partial v}, \quad \frac{\partial W'_q}{\partial v} = \frac{\partial \hat{W}'}{\partial v} + \sigma \cdot \frac{1}{2^n - 1} \text{round} \left( \frac{\hat{W}' - a}{s} \right) - \sigma \cdot \frac{\hat{W}' - a}{b - a}, \tag{8}$$

Here, $v$ denotes a trainable clipping bound, either the lower bound $a$ or the upper bound $b$. $\hat{W}' = \text{clip}(W', a, b)$ is the clipped weight matrix that governs how the quantizer adapts its effective range, and $\sigma$ is a sign factor that equals $-1$ when $v = a$ and $+1$ when $v = b$.

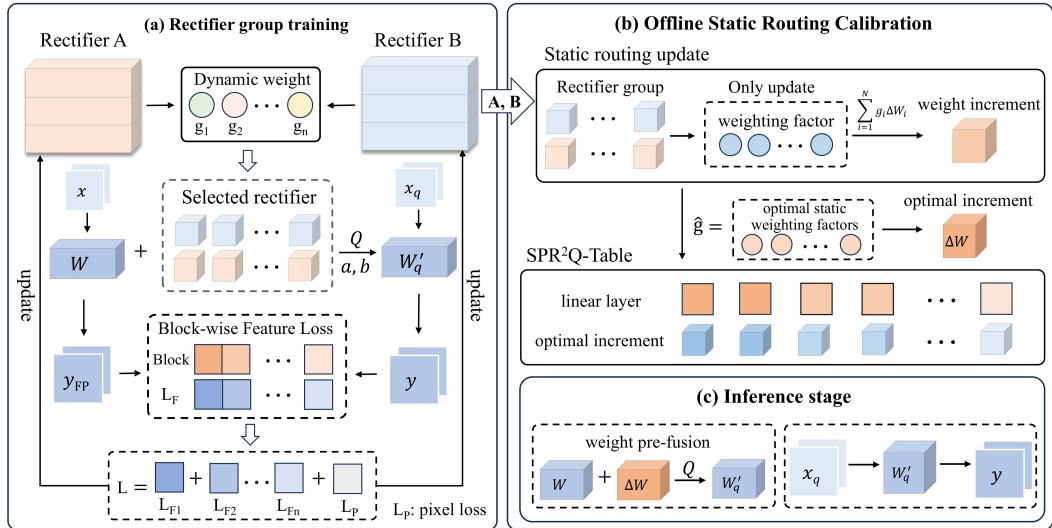

Figure 1: Overview of the SPR$^2$Q framework, showing its three stages: (a) Rectifier group Training, learning rectifiers with diverse complementary information via dynamic routing; (b) Offline Static Routing Calibration, generating the SPR$^2$Q Table to assign optimal increment for each layer; (c) Inference stage, performing computation using the updated and quantized weights.

Overall, this dual collaborative optimization enables two complementary effects: the rectifier $\Delta W$ learns to proactively counteract distortions introduced by quantization, while the clipping bounds $(a, b)$ dynamically refine the quantization mapping itself. After fine-tuning, the rectifier parameters are fused into the original weights, resulting in negligible inference overhead without altering the model's structure.

## 3.2 STATIC PRIORITY-BASED RECTIFIER ROUTING

To further enhance the model's quantization compensation capability and mitigate the homogenization issue caused by a single low-rank rectifier, we extend the single rectifier introduced in the previous section into an *rectifier group* composed of $N$ distinct rectifiers:

$$\mathcal{E} = \{\Delta W_1, \Delta W_2, \dots, \Delta W_N\}. \tag{9}$$

Within this mechanism, input information is routed to select the most suitable rectifier for augmentation, providing the model with a diverse set of alternative strategies for quantization compensation. Figure 1 illustrates the overall SPR$^2$Q framework and its three stages. Unlike traditional dynamic rectifier routing, which may introduce additional computational overhead and disrupt the original inference structure, we propose Static Priority-Based Rectifier Routing (SPR$^2$) module. In this framework, an offline evaluation stage pre-assigns the optimal, fixed rectifier to each component of the model. This design preserves the benefits of multiple rectifiers while avoiding extra inference cost and dynamic structural modifications.

**Rectifier group training**. To construct a group of $N$ distinct and high-performance rectifiers, we introduce a dynamic routing training stage. The goal of this stage is to encourage diverse rectifiers to be sufficiently engaged and optimized during training, enabling them to acquire specialized capabilities for handling heterogeneous information and compensating for different types of quantization errors.

Specifically, we employ a lightweight gating network that assigns input-dependent routing weights to each rectifier. Based on these weights, the increments $\Delta W_i$ produced by individual rectifiers are aggregated into a fused increment, which is then added to the base weights. The quantized effective weights used in the forward pass are given by:

$$\Delta W_i = B_i A_i, \quad W_q' = Q_{a,b}\left(W + \sum_{i=1}^{N} g_i \cdot \Delta W_i\right). \tag{10}$$

Here, each rectifier generates a rank-decomposed weight update $\Delta W_i$ through the product of its rectifier matrices $A_i$ and $B_i$. The gating network assigns a dynamic weight $g_i$ to each rectifier based on the input, and the weighted sum of all rectifier increments forms the fused update. This fused update is then added to the original weights $W$ and passed through the quantizer $Q_{a,b}$ to obtain the final quantized weights $W_q'$ used for inference.

With the effective weights $W_q'$ computed, the model output is obtained by a standard linear transformation of the input $X_q$:

$$Y = X_q W_q'. \tag{11}$$

During training, we minimize the hybrid loss function $\mathcal{L}$ (Eq. 6) to jointly optimize all rectifiers $\{(A_i, B_i)\}_{i=1}^N$. This strategy enables the rectifiers, under the guidance of the gating network, to learn input-dependent, specialized compensation. By doing so, each rectifier can handle different types of information, allowing diverse selections within the module to mitigate information loss caused by quantization. This not only enhances the model's representational capacity but also provides a range of compensation strategies, improving robustness to quantization errors.

**Offline Static Routing Calibration**. Following the Rectifier group training, we introduce the Offline Static Routing Calibration stage. The goal is to consolidate the diverse capabilities of the rectifiers learned during dynamic training into a fixed configuration. Specifically, we adopt the same loss function as used in the Rectifier group training stage, but with the pre-trained backbone weights and the learned rectifier parameters frozen. We employ a gradient descent-based method to calibrate and learn the weighting factors corresponding to each rectifier to obtain the optimal static weighting factors. Formally, given the permissible gating weight space $\mathcal{G}$, the optimization objective is:

$$\hat{g} = \arg\min_{g \in \mathcal{G}} \mathcal{L}\Big( f(X, Q_{a,b}(W + \sum_{i=1}^N g_i \Delta W_i)) \Big), \tag{12}$$

Here, $\hat{g}$ represents the optimal static weighting factors for combining multiple rectifiers, effectively capturing the diverse compensation strategies learned during dynamic training. The collected optimal static weighting factors are used to compute a weighted combination of the rectifier increments, resulting in the optimal increments, which are then organized to form the SPR$^2$Q Table shown in Figure 1.

**Inference stage**. Since the Offline Static Routing Calibration obtains the optimal increment for each module through the precomputed optimal gating weights, each module retrieves its corresponding optimal increment from the SPR$^2$Q Table and fuses it with the pretrained weights. The augmented weights are then quantized to produce the final weights used for forward computation. This design ensures that each module applies a fixed, optimal increment while preserving the original model structure, without requiring dynamic routing or introducing additional computational overhead.

## 4 EXPERIMENTS

### 4.1 EXPERIMENTAL SETTINGS

**Datasets and Evaluation.** In this work, we use DF2K (Agustsson & Timofte, 2017; Timofte et al., 2017) as the training set. This dataset consists of the DIV2K (Agustsson & Timofte, 2017) and Flickr2K (Timofte et al., 2017). We then employed five widely used benchmark datasets for evaluation: Set5 (Bevilacqua et al., 2012), Set14 (Zeyde et al., 2010), B100 (Martin et al., 2001), Urban100 (Huang et al., 2015), and Manga109 (Matsui et al., 2017). These are composed of 5, 14, 100, 100, and 109 images, respectively. In the benchmark evaluation, low-resolution inputs are fed into the quantization model for high-resolution image reconstruction, after which these reconstructed images are compared with the reference images. Performance is reported using PSNR and SSIM (Wang et al., 2004), measured on the Y channel of the YCbCr space.

**Training Details.** We adopt MambaIRv2-light (Guo et al., 2025) as the backbone and conduct experiments with scale factors of $\times 2$ and $\times 4$, evaluating all quantized models at 4-bit, and 2-bit precision. Hyperparameter settings are kept consistent across experiments. For optimization, we use the Adam optimizer (Kingma & Ba, 2015) with a learning rate of $1 \times 10^{-2}$ and $\beta = (0.9, 0.999)$, while the learning rate schedule follows a Cosine Annealing strategy (Loshchilov & Hutter, 2017)

Table 1: comparison with SOTA Mamba quantization methods on benchmark datasets for SR.

| Method | Bit | Set5(×2) | | Set14(×2) | | B100(×2) | | Urban100(×2) | | Manga109(×2) | |
|---|---|---|---|---|---|---|---|---|---|---|---|
| | | PSNR | SSIM | PSNR | SSIM | PSNR | SSIM | PSNR | SSIM | PSNR | SSIM |
| MambaIRv2-light | 32 | 38.26 | 0.9615 | 34.09 | 0.9221 | 32.36 | 0.9019 | 33.26 | 0.9378 | 39.35 | 0.9785 |
| PTQ4VM | 4 | 37.17 | 0.9549 | 32.86 | 0.9099 | 31.57 | 0.8900 | 30.47 | 0.9084 | 37.22 | 0.9706 |
| Quamba | 4 | 37.07 | 0.9544 | 32.77 | 0.9092 | 31.47 | 0.8896 | 30.54 | 0.9107 | 36.94 | 0.9699 |
| MambaQuant | 4 | 36.67 | 0.9495 | 31.76 | 0.8899 | 30.85 | 0.8756 | 28.08 | 0.8407 | 33.47 | 0.9186 |
| Ours (SPR$^2$Q) | 4 | **37.72** | **0.9589** | **33.27** | **0.9156** | **31.94** | **0.8964** | **31.53** | **0.9223** | **38.03** | **0.9754** |
| PTQ4VM | 2 | 34.38 | 0.9328 | 31.05 | 0.8886 | 30.21 | 0.8660 | 27.61 | 0.8603 | 32.04 | 0.9399 |
| Quamba | 2 | 34.66 | 0.9339 | 31.26 | 0.8899 | 30.38 | 0.8687 | 27.80 | 0.8613 | 32.50 | 0.9407 |
| MambaQuant | 2 | 34.65 | 0.9337 | 31.22 | 0.8885 | 30.36 | 0.8685 | 27.78 | 0.8610 | 32.43 | 0.9395 |
| Ours (SPR$^2$Q) | 2 | **35.97** | **0.9495** | **31.98** | **0.9020** | **30.95** | **0.8827** | **28.55** | **0.8819** | **34.39** | **0.9599** |
| Method | Bit | Set5(x4) | | Set14(x4) | | B100(x4) | | Urban100(x4) | | Manga109(x4) | |
| | | PSNR | SSIM | PSNR | SSIM | PSNR | SSIM | PSNR | SSIM | PSNR | SSIM |
| MambaIRv2-light | 32 | 32.51 | 0.8992 | 28.84 | 0.7878 | 27.75 | 0.7426 | 26.82 | 0.8079 | 31.24 | 0.9182 |
| PTQ4VM | 4 | 30.82 | 0.8670 | 27.69 | 0.7546 | 26.95 | 0.7115 | 24.76 | 0.7321 | 28.19 | 0.8660 |
| Quamba | 4 | 31.01 | 0.8715 | 27.77 | 0.7585 | 26.99 | 0.7149 | 25.01 | 0.7470 | 28.57 | 0.8752 |
| MambaQuant | 4 | 30.74 | 0.8650 | 27.17 | 0.7413 | 26.37 | 0.6920 | 23.28 | 0.6694 | 26.73 | 0.8186 |
| Ours (SPR$^2$Q) | 4 | **31.60** | **0.8844** | **28.27** | **0.7725** | **27.33** | **0.7274** | **25.64** | **0.7713** | **29.60** | **0.8959** |
| PTQ4VM | 2 | 28.77 | 0.8162 | 26.36 | 0.7167 | 26.16 | 0.6802 | 23.37 | 0.6704 | 25.26 | 0.7943 |
| Quamba | 2 | 28.88 | 0.8080 | 26.45 | 0.7131 | 26.20 | 0.6752 | 23.48 | 0.6651 | 25.43 | 0.7818 |
| MambaQuant | 2 | 28.84 | 0.8079 | 26.41 | 0.7114 | 26.18 | 0.6739 | 23.45 | 0.6648 | 25.38 | 0.7829 |
| Ours (SPR$^2$Q) | 2 | **29.37** | **0.8327** | **26.73** | **0.7319** | **26.42** | **0.6949** | **23.69** | **0.6874** | **25.77** | **0.8096** |

to ensure stable convergence. Specifically, the Rectifier group training is performed for 12,000 iterations, while the Offline Static Routing Calibration stage is conducted for 500 iterations. Both stages employ a batch size of 8. The rank of each rectifier module is set to $r = 8$. During the Rectifier group training stage, each group is configured with $N = 4$ parallel rectifiers. This is a trade-off we adopt to improve performance while maintaining training efficiency. This work is implemented based on the PaddlePaddle framework, and experiments are conducted on an NVIDIA RTX 4090 GPU.

## 4.2 COMPARISON WITH STATE-OF-THE-ART METHODS

We compare against PTQ4VM (Cho et al., 2025), Quamba (Chiang et al., 2025), and MambaQuant (Xu et al., 2025), which represent the strongest existing methods in the Mamba quantization literature. PTQ4VM is among the first methods specifically designed for post-training quantization of Visual Mamba. Quamba provides an effective baseline by combining quantization with architecture adaptation. MambaQuant employs variance-aligned rotation, effectively preserving performance across visual tasks—including image classification, object detection, and semantic segmentation—and language tasks. To enable a fair comparison, we report the performance of the full-precision MambaIRv2-light (Guo et al., 2025) model directly from the original paper. This is because none of these methods had previously been evaluated on the MambaIRv2-light super-resolution model. We applied them to the Mamba module within MambaIRv2-light, whilst all non-Mamba modules underwent uniform quantization using our method. This ensured all comparisons occurred within a consistent framework, enabling a fair assessment of performance variations arising from different quantization strategies.

**Quantitative results**. The table 1 presents a comprehensive comparison of various quantization methods at 4-bit and 2-bit depths, alongside scaling factors of ×2 and ×4. It can be observed that existing Mamba quantization methods, including PTQ4VM, Quamba, and MambaQuant, exhibit significant performance degradation when bit width is reduced, particularly on datasets rich in high-frequency details such as Urban100 and Manga109. For instance, PTQ4VM and MambaQuant show a marked decline in PSNR when transitioning from 4-bit to 2-bit quantization, highlighting their limited capacity to compensate for quantization errors in complex textured regions. In contrast, SPR$^2$Q consistently outperforms existing quantization methods across all evaluation scenarios. In the 4-bit precision test on the Set5 (×2) dataset, SPR$^2$Q achieves a PSNR value 0.55 dB higher than PTQ4VM and 1.05 dB higher than MambaQuant. More importantly, on the challenging Urban100 dataset, SPR$^2$Q outperforms existing baseline methods by approximately 1 dB in the 4-bit setting.

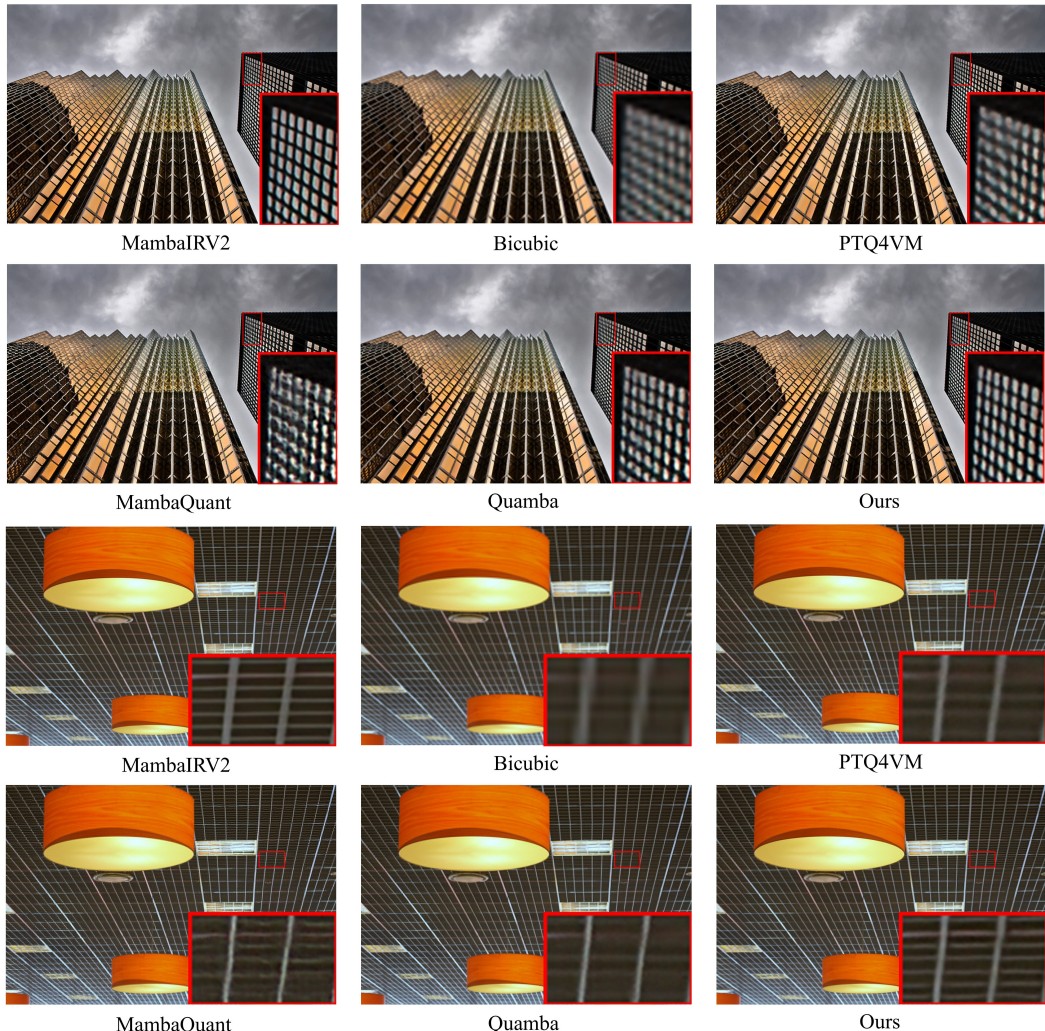

Figure 2: Visual comparison for image SR ($\times 4$) on Urban100(img019 and img044).

Even when precision is reduced to 2 bits, SPR$^2$Q maintains competitive performance, showing only a 1.75 dB degradation compared to its 4-bit counterpart on Set5 while achieving a significant 1.31 dB improvement over other state-of-the-art methods.

These results demonstrate the effectiveness of the rectified group and static priority routing mechanism in mitigating quantization performance degradation. Meanwhile, SPR$^2$Q demonstrates strong performance across different datasets and scaling factors, highlighting its robustness in handling diverse texture distributions and complex scenarios.

**Qualitative results**.We present the visual comparison results for $\times 4$ (see Figure 2). It can be observed that the three contrast-based quantization methods exhibit significant shortcomings in detail recovery. The images appear blurred overall, with severe loss of texture and fine structure, and edges often show diffusion and misalignment. Our method restores texture and edge details more clearly while preserving the overall structure, enabling the images to present richer high-frequency information.

### 4.3 ABLATION STUDY

**Impact of Component Modules**. The effect of different modules on performance is analyzed in Table 2a. First, the Pre-Quantization Fine-tuning with Fused Rectifier (PQFR) utilizes a single set of learnable low-rank parameters to proactively compensate for quantization errors. Introducing

Table 2: Ablation studies. Models are trained on DF2K, and tested on Set5 (x2) and Urban100 (x2).

| PQFR | RGT | OSRC | Set5 PSNR | Set5 SSIM | Urban100 PSNR | Urban100 SSIM |
|------|-----|------|------|------|------|------|
| | | | 37.20 | 0.9554 | 30.69 | 0.9112 |
| ✓ | | | 37.44 | 0.9567 | 31.25 | 0.9188 |
| ✓ | ✓ | | 37.60 | 0.9581 | 31.24 | 0.9170 |
| ✓ | ✓ | ✓ | 37.72 | 0.9589 | 31.53 | 0.9223 |

(a) Impact of component modules

| $r$ | Set5 PSNR | Set5 SSIM | Urban100 PSNR | Urban100 SSIM |
|-----|------|------|------|------|
| 2 | 37.63 | 0.9585 | 31.36 | 0.9208 |
| 4 | 37.67 | 0.9588 | 31.48 | 0.9221 |
| 8 | 37.72 | 0.9589 | 31.53 | 0.9223 |
| 16 | 37.74 | 0.9591 | 31.70 | 0.9240 |

(b) Rectifier rank

Table 3: Rectifier group size

| size | Set5 (x2) PSNR | Set5 (x2) SSIM | Urban100 (x2) PSNR | Urban100 (x2) SSIM |
|------|------|------|------|------|
| 2 | 37.50 | 0.9578 | 31.31 | 0.9196 |
| 4 | 37.72 | 0.9589 | 31.56 | 0.9223 |
| 8 | 37.82 | 0.9595 | 31.73 | 0.9249 |

Table 4: Efficiency on MambaIRv2-light

| Metric | FP32 | 4bit | 2bit |
|--------|------|------|------|
| Model Size (MB) | 3.01 | 1.20 | 1.07 |
| Compression Ratio | 1.00× | 2.51× | 2.81× |
| FLOPs (G) | 75.6 | 22.0 | 18.2 |
| Speed-up | 1.00× | 3.44× | 4.15× |

only the PQFR module already improved the baseline by 0.24 dB on the Set5 dataset and by 0.56 dB on the Urban100 dataset. This demonstrates that fusing learnable rectifiers prior to quantization successfully injects a compensation mechanism into the backbone network, significantly mitigating the information loss caused by discretisation. Next, we employ Rectifier Group Training (RGT) to extend the single low-rank parameter set into multiple groups for fusion. Upon further enabling RGT, performance improved by an additional 0.16 dB on the Set5 dataset. This demonstrates that using diverse experts effectively expands the representation space. Finally, the Offline Static Routing Calibration (OSRC) is incorporated to calibrate the weighting factors of these multiple low-rank parameter groups. Enabling OSRC yields a further gain of 0.12 dB on Set5. This confirms that refining the fusion weights leads to optimal information compensation.

**Rectifier Rank**. We further examine the effect of the rectifier rank $r$ on performance, as shown in Table 2b. The results indicate that reconstruction quality consistently improves as the rank increases from 2 to 8. However, the performance gain begins to saturate beyond this point; for instance, increasing $r$ to 16 yields only a marginal 0.02 dB improvement on Set5 but entails higher training parameter costs. Consequently, we select $r = 8$ as the default setting, as the performance improvement becomes negligible beyond this point, indicating that a rank of 8 is sufficient for effective quantization compensation.

**Rectifier Group Size**. Table 3 reports the impact of varying the rectifier group size. Expanding the group size from 2 to 4 yields significant PSNR improvements of 0.22 dB on Set5 and 0.25 dB on Urban100, demonstrating that a larger group effectively enhances the model's ability to select optimal rectifier paths and improves its representational capacity. Increasing the group size to 8, however, only brings marginal gains of 0.10 dB (Set5) and 0.17 dB (Urban100), considerably smaller than the improvement from 2 to 4. This indicates that a group size of 4 is sufficient to capture the diverse compensation patterns required for quantization. Therefore, we adopt $N = 4$ as the default setting, representing the saturation point where the model achieves robust compensation capability.

## 4.4 PRACTICAL EFFICIENCY OF SPR$^2$Q

To validate the practical efficiency of SPR$^2$Q, we evaluate the model size, computational cost (FLOPs), and inference acceleration compared to the full-precision baseline. All statistics are collected based on the MambaIRv2-light model under the ×4 super-resolution setting, using an input resolution of (180, 320). As shown in Table 4, SPR$^2$Q can compress the model to 4 and 2 bits with the compression ratio being 2.51× and 2.81×, and speedup ratio being 3.44× and 4.15×. Crucially, these gains are achieved with zero additional inference cost, as all auxiliary parameters are fused offline.

Table 5: Quantitative comparison with SOTA methods on SwinIR-light.

| Method | Bit | Set5(x2) | | Set14(x2) | | B100(x2) | | Urban100(x2) | | Manga109(x2) | |
|---|---|---|---|---|---|---|---|---|---|---|---|
| | | PSNR | SSIM | PSNR | SSIM | PSNR | SSIM | PSNR | SSIM | PSNR | SSIM |
| SwinIR-light | 32 | 38.14 | 0.9611 | 33.86 | 0.9206 | 32.31 | 0.9012 | 32.76 | 0.9340 | 39.12 | 0.9783 |
| 2DQuant | 2 | 36.00 | 0.9497 | 31.98 | 0.9012 | 30.91 | 0.8810 | 28.62 | 0.8819 | 34.40 | 0.9602 |
| FIMA-Q | 2 | 36.06 | 0.9515 | 32.10 | 0.9048 | 31.01 | 0.8848 | 28.77 | 0.8873 | 34.75 | 0.9638 |
| APHQ-ViT | 2 | 36.14 | 0.9517 | 32.14 | 0.9049 | 31.04 | 0.8850 | 28.86 | 0.8885 | 34.99 | 0.9644 |
| **Ours (SPR$^2$Q)** | 2 | **37.28** | **0.9572** | **32.83** | **0.9113** | **31.63** | **0.8921** | **30.20** | **0.9077** | **37.08** | **0.9726** |

Table 6: Exploration of our SPR$^2$Q method under 1-bit quantization.

| **Method** | scale | Set5 | | Set14 | | B100 | | Urban100 | | Manga109 | |
|---|---|---|---|---|---|---|---|---|---|---|---|
| | | PSNR | SSIM | PSNR | SSIM | PSNR | SSIM | PSNR | SSIM | PSNR | SSIM |
| Ours (SPR$^2$Q) | ×2 | 34.82 | 0.9428 | 31.27 | 0.8956 | 30.41 | 0.8754 | 27.76 | 0.8690 | 32.38 | 0.9505 |
| Ours (SPR$^2$Q) | ×4 | 28.84 | 0.8213 | 26.41 | 0.7215 | 26.21 | 0.6852 | 23.41 | 0.6751 | 25.31 | 0.7995 |

## 4.5 CROSS-ARCHITECTURE GENERALIZATION

We evaluate SPR$^2$Q under extreme low-bit settings on the Transformer-based SwinIR-light (Liang et al., 2021) and compare it with representative SOTA Vision Transformer quantization methods: 2DQuant (Liu et al., 2024), FIMA-Q (Wu et al., 2025a), and APHQ-ViT (Wu et al., 2025b). All experiments are conducted under a unified DF2K setting. For FIMA-Q and APHQ-ViT, we align our Rectifier Group Training with their Quantization Reconstruction phase (Batch Size = 16, iterations = 4000); for 2DQuant, we adopt its original settings (Batch Size = 32, iterations = 3000).

Under 2-bit quantization for ×2 super-resolution (Table 5), SPR$^2$Q outperforms all compared baselines, improving PSNR by 1.14 dB on Set5, with a substantial gain of 1.34 dB on the texture-rich Urban100 dataset. These results indicate that SPR$^2$Q is highly effective at preserving high-frequency details even under extreme low-bit compression, and its performance is robust across different model architectures.

## 4.6 EXTREME 1-BIT QUANTIZATION

Evaluation of SPR$^2$Q on MambaIRv2-light under extreme 1-bit quantization is presented in Table 6. For ×2 scaling, the model achieves a PSNR of 34.82 dB, and for ×4 scaling, 28.84 dB. Compared to the 2-bit results, the performance drop is moderate, demonstrating that SPR$^2$Q remains effective in preserving reconstruction quality even under extreme quantization.

## 5 CONCLUSION

In this work, we advance the study of low-bit quantization for super-resolution models built on the Mamba architecture. We first identify that existing Mamba quantization methods exhibit significant domain adaptation issues under low-bit SR settings. To address this, we propose SPR$^2$Q, a quantization framework specifically designed for low-bit SR. SPR$^2$Q employs rectifiers to compensate for information loss introduced by quantization and jointly optimizes both the rectifier and quantizer parameters, enabling the model to adapt effectively to the quantization process. Moreover, we introduce the Static Priority-Based Rectifier Routing mechanism to provide diverse compensation strategies and calibrate a static routing table, allowing the model to efficiently obtain optimal increments from the rectifier group during inference. This design preserves the original model structure while incurring negligible additional computational overhead. Extensive experiments demonstrate that SPR$^2$Q consistently outperforms existing Mamba SOTA quantization methods across various low-bit settings and exhibits strong cross-model generalization. It substantially improves reconstruction quality and detail fidelity, providing a novel and effective solution for low-bit SR quantization.

ACKNOWLEDGMENTS

This work was supported in part by the National Natural Science Foundation of China under Grant U22A2096 and Grant 62206211; in part by the Scientific and Technological Innovation Teams in Shaanxi Province under Grant 2025RS-CXTD-011; in part by Shaanxi Province Core Technology Research and Development Project under Grant 2024QY2-GJHX-11; in part by the CCF-Kuaishou Large Model Explorer Fund under Grant CCF-KuaiShou 2025006; and in part by the Young Talent Fund of Association for Science and Technology, Shaanxi, China, under Grant 20240140.

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
