# OpenReview forum: "SPR$^2$Q: Static Priority-based Rectifier Routing Quantization for Image Super-Resolution"
_ICLR.cc/2026/Conference — ICLR 2026 Poster_

### Official Review · Reviewer_xceu · 2025-10-27

**Soundness:** 3
**Presentation:** 3
**Contribution:** 3
**Rating:** 4
**Confidence:** 5

**Summary:**

This paper proposes SPR2Q, a static priority-based rectifier routing quantization method. The overall pipeline can be divided into two parts. The first part leverages a low-rank rectifier group to enhance the backbond network. The second part uses offline static routing calibration to obtain the SPR2Q table to assign the optimal increment for each layer. The proposed method achieves SOTA performance on Mamba with five commonly used benchmarks.

**Strengths:**

- The design is effective and clear, obtaining SOTA performance with both metric and visual comparison on the SR task.
- The writing is clear and easy to follow.

**Weaknesses:**

- The proposed method is only tested on MambaIRv2-light. However, the proposed method can be safely tested on more models, including MambaIRv2_SR2, SwinIR. Please provide the results on these models to demonstrate the generalization ability.
- Please provide the complexity of the SPR2Q model, including model parameters, storage, inference speedup ratio, and GPU memory usage. These metrics are critical to the model lighting.
- The rank of each recitifier module is important. However, the influence of the rank is not discussed in the ablation study. Please provide the results of various ranks and the results of full-rank.

**Questions:**

- In Figure 1, it should be "update" instead of "updata". The corresponding
- What is the limit of the contribution from the Rectifier group size?

---

> ### Author Response · Authors · 2025-11-24
> **Response to Reviewer xceu**
>
> We thank you for the positive comments on our method and writing clarity, and we appreciate the constructive suggestions on generalization, complexity analysis, and rank ablations, which help us further improve the quality of the paper.
>
> **Q1:Generalization on MambaIRv2-B and SwinIR.**
>
> A: To validate the generality of SPR²Q, we expanded our evaluation to both larger-scale and cross-architecture settings.
> 1. Large-scale Model (MambaIRv2-B): We applied SPR²Q to the 22.9M-parameter MambaIRv2-B (vs. 0.77M for Light). With batch size adjusted from 8 to 1, the method still delivers strong 4-bit performance.
>
> |Method|Bit|Set5(×2)||Set14(×2)||B100(×2)|| Urban100(×2)||Manga109(×2)||
> |-|-|-|-|-|-|-|-|-|-|-|-|
> |||PSNR|SSIM|PSNR|SSIM|PSNR|SSIM|PSNR|SSIM| PSNR| SSIM|
> |MambaIRv2-B|32|38.65|0.9631|34.89|0.9275|32.62|0.9053|34.49|0.9468|40.42|0.9810|
> |PTQ4VM|4|37.36|0.9569|33.07|0.9131|31.72|0.8926|31.01|0.9155|37.54|0.9734|
> |SPR²Q (Ours)|4|37.97|0.9599|33.78|0.9196|32.19|0.8999|32.69 |0.9340|38.73|0.9772|
>
> On the larger MambaIRv2-B architecture, SPR²Q maintains high fidelity under 4-bit quantization, exhibiting only a marginal performance drop (0.68 dB on Set5) compared to the FP32 baseline.
>
> 2. Cross-Architecture (SwinIR-light):
> We additionally tested SPR²Q on the Transformer-based SwinIR-light under ×2 SR. These results confirm that SPR²Q applies reliably to heterogeneous SR backbones.
>
> |Method|Bit|Set5(×2)||Set14(×2)||B100(×2)||Urban100(×2)||Manga109(×2)| |
> |-|-|-|-|-|-|-|-|-|-|-|-|
> |||PSNR|SSIM|PSNR|SSIM|PSNR|SSIM|PSNR|SSIM|PSNR|SSIM |
> | SwinIR-light|32|38.14|0.9611|33.86|0.9206|32.31|0.9012|32.76|0.9340|39.12|0.9783|
> |Ours(SPR²Q)|4|37.95|0.9602|33.55|0.9178|32.14|0.8989|32.04|0.9275|38.59|0.9769|
> |Ours(SPR²Q)|2|37.28|0.9572|32.83|0.9113|31.63|0.8921|30.20|0.9077|37.08|0.9726|
>
> SPR²Q consistently maintains high performance under both 4-bit and 2-bit quantization settings, demonstrating strong generalization capability beyond MambaIRv2-light. These results on SwinIR-light confirm that SPR²Q is architecture-agnostic.
>
> **Q2:Model complexity analysis.**
>
> A:To demonstrate the efficiency of SPR²Q, we report the complexity metrics on MambaIRv2-light (input $180 \times 320$, $\times 4$ SR). Regarding the requested GPU memory usage, since our current implementation utilizes pseudo-quantization (simulating low-bit operations via FP32), direct runtime memory measurements do not accurately reflect the hardware savings of actual low-bit deployment. Therefore, we report Model Size (Storage) as a direct proxy for static memory footprint, where the reduction implies a proportional decrease in memory usage on deployed devices. Crucially, thanks to our offline fusion strategy, the method achieves significant storage reduction and inference acceleration without introducing any additional structural overhead.
>
> |Metric|FP32|4bit|2bit|
> |-|-|-|-|
> |Model Size (MB)|3.01|1.20|1.07|
> |Compression Ratio|1.00×|2.51×|2.81×|
> |FLOPs (G)|75.6|22.0|18.2|
> |Speed-up|1.00×| 3.44×|4.15×|
>
> As shown in the table, the quantized model maintains a parameter count (792K) virtually identical to the FP32 baseline (790K), with the negligible increase solely attributable to quantization scale parameters. SPR²Q can compress the model to 4 and 2 bits with the compression ratio being 2.51× and 2.81×, and speedup ratio being 3.44× and 4.15×. Benefiting from the Offline Static Routing mechanism, SPR²Q achieves the theoretical upper limit of compression and speedup while maintaining superior performance.
>
> **Q3:influence of the rank.**
>
> A:We have conducted an ablation study on the rank $r$ of the rectifier module, including a comparison with the Full-Rank setting. The results are reported below:
>
> |r|Set5(×2)||Urban100(×2)||
> |-|-|-|-|-|
> ||PSNR|SSIM|PSNR|SSIM|
> |2|37.63|0.9585|31.36|0.9208|
> |4|37.67|0.9588|31.48|0.9221|
> |8|37.72|0.9589|31.53|0.9223|
> |16|37.74|0.9591|31.70|0.9240|
> |full-rank|37.73|0.9592|31.52|0.9229|
>
> The results indicate that performance improves with rank but saturates noticeably beyond $r=8$. Furthermore, the Full-Rank setting yields comparable performance to the low-rank settings ($r=16$).
>
> **Q4:Writeup revision.**
>
> A:We thank you for spotting this typo. We will correct “updata” to “update” in Figure 1 and carefully checked the manuscript to avoid similar issues.
>
> **Q5:Limit of the contribution from the Rectifier group size.**
>
> A:The contribution of the Rectifier group size shows a clear saturation trend. Increasing the size from 2→4 brings a noticeable gain, while enlarging it further to 8 provides only marginal improvement. Although we initially selected N = 4 due to training-cost considerations, our additional tests show that larger group sizes do not increase memory usage. Therefore, the practical limit is determined by performance saturation rather than computational constraints, and N = 4 already captures most of the achievable benefit.

---

> > ### Comment · Reviewer_xceu · 2025-11-24
> >
> > Thanks the authors for the detailed rebuttal. The rebuttal has addressed my concerns and I decide to raise my score to 6.

---

### Official Review · Reviewer_xpnk · 2025-10-29

**Soundness:** 3
**Presentation:** 3
**Contribution:** 3
**Rating:** 6
**Confidence:** 5

**Summary:**

This paper introduces Static Priority-based Rectifier Routing Quantization (SPR2Q), a novel post-training quantization method for low-bit image super-resolution. Unlike existing approaches that struggle with severe information loss under extreme low-bit settings, SPR2Q injects rich compensation information into the model before quantization by embedding a low-rank rectifier group during fine-tuning. A static rectifier priority routing mechanism then evaluates each rectifier’s capability offline and updates weights accordingly without adding inference overhead.

**Strengths:**

The idea of learning compensation information through lightweight rectifiers is very interesting. In addition, the use of dynamic routing during training followed by static rectifier routing is an effective design choice, improving performance without increasing computational cost. Experimental results support the effectiveness of the proposed method.

**Weaknesses:**

1. The method is tested only on a single SR baseline, MambaIRv2-light. To better demonstrate its applicability and generalization capability, it should also be tested against other baselines, such as MaIR (CVPR 2025), EAMamba (ICCV 2025), and First-Order State Space Model for Lightweight Image Super-Resolution (ICASSP 2025).

2. Since the main contribution lies in the use of multiple rectifiers, a more in-depth analysis of their internal behavior is expected. For example, the authors mention that each rectifier handles different types of information, but visual or statistical evidence is needed to support this claim. In addition, the analysis of optimal gate weights should be expanded to better explain their optimality.

3. The performance of PTQ depends heavily on the training dataset. In particular, the proposed offline routing calibration method appears to target different domains; however, the experiments are conducted using only a single training dataset.

4. The paper would benefit from careful linguistic revision and thorough proofreading.

**Questions:**

The authors used rectifiers of the same size. Have the authors considered using rectifiers with different capacities, so that multiple experts can more effectively distribute the workload?

---

> ### Author Response · Authors · 2025-11-24
> **Response to Reviewer xpnk**
>
> We thank you for the positive remarks on our design and experimental results, and we appreciate the constructive suggestions regarding generalization, analysis depth, and presentation, which help us further refine the paper.
>
> **Q1:Demonstrate applicability and generalization capability.**
>
> A:Following the reviewers’ collective suggestions, we additionally evaluated SPR²Q on a Transformer-based baseline (SwinIR-light). As shown in the new results below:
> |Method|Bit|Set5(×2)||Set14(×2)||B100(×2)||Urban100(×2)||Manga109(×2)||
> |-|-|-|-|-|-|-|-|-|-|-|-|
> |||PSNR|SSIM|PSNR|SSIM|PSNR|SSIM|PSNR|SSIM|PSNR|SSIM|
> |SwinIR-light|32|38.14|0.9611|33.86|0.9206|32.31|0.9012|32.76|0.9340|39.12|0.9783|
> |Ours(SPR²Q)|4|37.95|0.9602|33.55|0.9178|32.14|0.8989|32.04|0.9275|38.59|0.9769|
> |Ours(SPR²Q)|2|37.28|0.9572|32.83|0.9113|31.63|0.8921|30.20|0.9077|37.08|0.9726|
>
> SPR²Q consistently improves both 4-bit and 2-bit quantization performance, demonstrating strong generalization beyond MambaIRv2-light.The results on SwinIR-light confirm that SPR²Q is architecture-agnostic.
>
> **Q2:Analysis of Rectifier Behavior and the Optimality of Learned Gating Weights.**
>
> A:1.To verify that the multiple rectifiers learn different types of information, we analyzed a representative intermediate layer using only the incremental outputs. Inspired by Xie et al., ICCV 2021 (“Learning Frequency-aware Dynamic Network for Efficient Super-Resolution”), which highlights that low-frequency regions typically correspond to smooth areas with less texture, while high-frequency regions correspond to edges and complex textures, we quantified each expert’s spectral preference. Specifically, we performed spectrum analysis on the feature increments generated by each rectifier and calculated the ratio of high-frequency to low-frequency energy. This allows us to distinguish between rectifiers that preferentially process texture-oriented (high-frequency) or structure-oriented (low-frequency) information.
>
> In Layer, the rectifiers show clear specialization:
> |Rectifier |Gating| HF/LF Ratio| Low Energy| High Energy|
> |-|-|-|-|-|
> |0|0.232|1.5022|-2.5575|-3.8419|
> |1|0.222|3.2094|-1.1601|-3.7232|
> |2|0.323|3.2506|-0.8874|-2.8845|
> |3|0.223|2.2088|-1.7173|-3.7932|
>
> Rectifiers 1 and 2 have much higher HF/LF ratios (~3.2), meaning they focus on textures and high-frequency details. In contrast, Rectifier 0 has a much lower ratio (1.50), indicating a preference for smooth, low-frequency structures. Notably, Expert 2 also has the highest output energy and gating weight, indicating that the model shows a clear preference for the detail-oriented rectifier at this layer to enhance reconstruction quality.
>
> 2.To validate that the learned gating weights are indeed optimal, we add an ablation comparing the model with and without Offline Static Routing Calibration, which refines the expert routing based on the learned gates. As shown below, enabling this module consistently improves performance:
>
> |Offline Static Routing Calibration|Set5(×2)||Urban100(×2)||
> |-|-|-|-|-|
> ||PSNR|SSIM|PSNR|SSIM|
> |✗|37.60|0.9581|31.24|0.9170|
> |✓|37.72|0.9589|31.53|0.9223|
>
> These gains confirm that the learned gating weights lead to better expert selection and are therefore optimal.
>
> **Q3:Dependence on training dataset and cross-domain generalization.**
>
> A:We appreciate the reviewer’s concern regarding the reliance on a single dataset. Our setup follows standard SR protocols, and the results demonstrate strong generalization across diverse domains. Although the offline routing calibration uses DF2K (natural images), we evaluated SPR²Q on benchmark datasets with distinct distributions: Manga109 (comics/cartoons with flat regions and sharp synthetic lines) and Urban100 (high-frequency repetitive textures). As shown by the experimental results in our paper, SPR²Q achieves state-of-the-art performance on these out-of-domain datasets, including 4-bit ×2 SR. This confirms that the offline routing calibration does not overfit DF2K but instead learns robust compensation patterns that generalize effectively to unseen domains.
>
> **Q4:Linguistic revision.**
>
> A:We thank you for your suggestion regarding language and presentation. We will carefully proofread and thoroughly revise the manuscript to ensure clarity, accuracy, and smoothness of expression.
>
> **Q5:Using rectifiers with different capacities.**
>
> A:Regarding your question about using rectifiers with different capacities, we also conducted an experiment where the four rectifiers were assigned distinct ranks (r=2, 4, 8, 16). The results are shown below:
> |Set5(×2)||Urban100(×2)||
> |-|-|-|-|
> |PSNR|SSIM|PSNR|SSIM|
> |37.74|0.9591|31.58|0.9233|
>
> Compared to the homogeneous baseline reported in the paper (where all 4 rectifiers are set to r=8), this heterogeneous configuration yields marginal improvements. The results indicate that there is no significant performance difference between the two settings.

---

> > ### Comment · Reviewer_xpnk · 2025-11-26
> > **RE:**
> >
> > Thanks to the authors for addressing my comments, which clarified most of my concerns. Regarding the first comment, I understand that the authors could not evaluate the method against the stronger SR baselines I mentioned. However, since the proposed method is not compared with any other quantization approaches in this experiment, it remains questionable whether “SPR²Q consistently improves both 4-bit and 2-bit quantization performance".

---

> > > ### Author Response · Authors · 2025-12-02
> > > **Response to Reviewer xpnk**
> > >
> > > We thank you for the further feedback. We acknowledge the concern regarding the lack of comparisons with other quantization schemes. To convincingly demonstrate the superiority of SPR²Q, we have incorporated comparisons with representative state-of-the-art (SOTA) methods tailored for Vision Transformers, including 2DQuant (NeurIPS 2024), a specialized scheme for SwinIR, as well as FIMA-Q (CVPR 2025) and APHQ-ViT (CVPR 2025), which represent the latest advancements in Vision Transformer quantization.
> > >
> > > To ensure a strictly fair comparison, we re-conducted the experiments under a unified training and evaluation configuration on the DF2K dataset. Specifically, for FIMA-Q and APHQ-ViT, we aligned the settings of our Rectifier Group Training phase with their Quantization Reconstruction phase (Batch Size = 16, Iterations = 4000), while for 2DQuant, we adopted the settings recommended in its original paper (Batch Size = 32, Iterations = 3000). In addition, we have revised the previously reported results to reflect this unified experimental protocol.
> > >
> > > The comparative results under the challenging 2-bit setting are presented below:
> > > |Method|Bit|Set5(×2)||Set14(×2)||B100(×2)||Urban100(×2)||Manga109(×2)||
> > > |-|-|-|-|-|-|-|-|-|-|-|-|
> > > |||PSNR|SSIM|PSNR|SSIM|PSNR|SSIM|PSNR|SSIM|PSNR|SSIM|
> > > |SwinIR-light|32|38.14|0.9611|33.86|0.9206|32.31|0.9012|32.76|0.9340|39.12|0.9783|
> > > |2Dquant|2|36.00|0.9497|31.98|0.9012|30.91|0.8810|28.62|0.8819|34.40|0.9602|
> > > |FIMA-Q|2|36.06|0.9515|32.10|0.9048|31.01|0.8848|28.77|0.8873|34.75|0.9638|
> > > |APHQ-ViT|2|36.14|0.9517|32.14|0.9049|31.04|0.8850|28.86|0.8885|34.99|0.9644|
> > > |Ours(SPR²Q)|2|37.28|0.9572|32.83|0.9113|31.63|0.8921|30.20|0.9077|37.08|0.9726|
> > >
> > > |Method|Bit|Set5(×4)||Set14(×4)||B100(×4)||Urban100(×4)||Manga109(×4)||
> > > |-|-|-|-|-|-|-|-|-|-|-|-|
> > > |||PSNR|SSIM|PSNR|SSIM|PSNR|SSIM|PSNR|SSIM|PSNR|SSIM|
> > > |SwinIR-light|32|32.44|0.8976| 28.77|0.7858|27.69|0.7406|26.47|0.7980|30.92|0.9151|
> > > |2Dquant|2|29.53|0.8372|26.86|0.7322|26.46|0.6927|23.84|0.6912|26.07|0.8163|
> > > |FIMA-Q|2|30.06|0.8529|27.21|0.7462|26.69|0.7057|24.17|0.7118|26.80|0.8403|
> > > |APHQ-ViT|2|30.12|0.8543|27.27|0.7472|26.71|0.7061|24.23|0.7143|27.00|0.8441|
> > > |Ours(SPR²Q)|2|31.04|0.8760|27.87|0.7632|27.10|0.7193|24.87|0.7420|28.38|0.8759|
> > >
> > > The results demonstrate that SPR²Q achieves state-of-the-art performance across all datasets and scales. Under 2-bit quantization for ×2 super-resolution, compared to the strongest baseline (APHQ-ViT), SPR²Q improves PSNR by 1.14 dB on Set5, with a substantial gain of 1.34 dB on the texture-rich Urban100 dataset. The improvement remains significant at the challenging ×4 scale, with SPR²Q outperforming APHQ-ViT by 0.92 dB on Set5 and 0.64 dB on Urban100. These results confirm that our method is highly robust and effective even under extreme low-bit compression.
> > >
> > > To ensure reproducibility, we provide the anonymous code repository at: https://github.com/momo5-a11/SPR2Q.
> > >
> > > References:
> > >
> > > [1] Kai Liu, Haotong Qin, Yong Guo, Xin Yuan, Linghe Kong, Guihai Chen and Yulun Zhang, “2DQuant: Low-bit Post-Training Quantization for Image Super-Resolution”, Advances in Neural Information Processing Systems (NeurIPS) 37, 2024.
> > >
> > > [2] Zhuguanyu Wu, Shihe Wang, Jiayi Zhang, Jiaxin Chen and Yunhong Wang, “FIMA-Q: Post-Training Quantization for Vision Transformers by Fisher Information Matrix Approximation”, Proceedings of the IEEE/CVF Conference on Computer Vision and Pattern Recognition (CVPR) 2025, pp. 14891-14900.
> > >
> > > [3] Zhuguanyu Wu, Jiayi Zhang, Jiaxin Chen, Jinyang Guo, Di Huang and Yunhong Wang, “APHQ-ViT: Post-Training Quantization with Average Perturbation Hessian Based Reconstruction for Vision Transformers”, Proceedings of the IEEE/CVF Conference on Computer Vision and Pattern Recognition (CVPR) 2025, pp. 9686-9695.

---

### Official Review · Reviewer_bGTK · 2025-10-30

**Soundness:** 3
**Presentation:** 2
**Contribution:** 3
**Rating:** 6
**Confidence:** 4

**Summary:**

This paper proposes SPR2Q, a post-training quantization (PTQ) framework specifically designed for image super-resolution (SR) with an emphasis on Mamba-based architectures. The key insight is that current PTQ techniques do not sufficiently prepare the model for quantization, especially under aggressive bit-width reduction (e.g., 2-bit or 1-bit).

To address this, the authors propose two complementary techniques:
1.	Pre-Quantization Fine-Tuning with Fused Rectifiers (PQFR):
Learnable, low-rank rectifiers are fused into model weights before quantization to reduce error and preserve representational fidelity.
2.	Static Priority-Based Rectifier Routing (SPR2):
A mechanism for statically assigning rectifiers to layers using an offline-calibrated routing table, which introduces diversity without runtime overhead.

Extensive experiments on five standard SR datasets and comparisons against three strong Mamba-specific quantization baselines (PTQ4VM, Quamba, and MambaQuant) show consistent performance improvements, especially in low-bit regimes.

**Strengths:**

1.	The paper clearly identifies a relevant and under-addressed problem: the difficulty of applying post-training quantization to super-resolution models, especially in architectures like Mamba that are sensitive to numerical errors due to their recurrent components. The need for specialized approaches in SR is well-motivated.
2.	The PQFR module is inspired by LoRA but adapted for PTQ in the SR setting, which is novel. The SPR2 routing table design is simple, efficient, and effective. It offers a practical compromise between dynamic routing (high cost) and naive shared rectifiers (low performance). And the method is well-integrated into existing training pipelines and avoids runtime penalties.
3.	The method requires only modest fine-tuning and has negligible inference overhead. It is applicable in edge scenarios where low-bit inference is essential.

**Weaknesses:**

1.	The entire evaluation is performed on the MambaIRv2-light model. While the motivation is grounded in the Mamba architecture, the method itself (particularly PQFR) should generalize. Add at least one experiment on a Transformer-based SR model (e.g., SwinIR) or a CNN-based model (e.g., EDSR) to support claims of generality.
2.	The routing table optimization (Equation 12) is essential to SPR2Q, yet the paper omits important implementation details. It is unclear whether the optimization of gating weights is performed via exhaustive search, gradient descent, or heuristics. I would like to see more information about how the routing weights ĝ are optimized, how long this process takes, and whether it scales with model size.
3.	Although the paper claims no additional inference cost, no quantitative measurements are presented. This is important to validate the claim that the fused rectifiers do not affect runtime. I would be great if include a table comparing runtime, model size, and memory usage (before and after SPR2Q), even on a single dataset, to support the efficiency claim.

**Questions:**

See the weakness

---

> ### Author Response · Authors · 2025-11-24
> **Response to Reviewer bGTK**
>
> We thank you for the encouraging assessment of our work, especially the recognition of its novelty, practical design, and relevance to SR quantization. We also appreciate the constructive suggestions on evaluation breadth and implementation details, which help us further improve the paper.
>
> **Q1: Generalization beyond MambaIRv2-light**
>
> A: We extended our evaluation to a Transformer-based SR model, SwinIR-light, to further verify the generality of SPR²Q. We conducted experiments under the ×2 setting with both 4-bit and 2-bit quantization. The results are summarized below:
> |Method|Bit|Set5(×2)||Set14(×2)  |  | B100(×2)|  | Urban100(×2)  | | Manga109(×2)| |
> |-|-|-|-|-|-|-|-|-|-|-|-|
> ||| PSNR| SSIM|PSNR| SSIM | PSNR | SSIM| PSNR  | SSIM | PSNR| SSIM |
> | SwinIR-light    | 32  | 38.14|0.9611|33.86 | 0.9206 | 32.31| 0.9012 | 32.76 | 0.9340 |  39.12 | 0.9783|
> |Ours(SPR²Q)|4|37.95|0.9602|33.55|0.9178|32.14|0.8989|32.04|0.9275|38.59|0.9769|
> |Ours(SPR²Q)|2|37.28|0.9572|32.83|0.9113|31.63|0.8921|30.20|0.9077|37.08|0.9726|
>
> The results on SwinIR-light compellingly demonstrate the exceptional universality of SPR²Q. It seamlessly adapts to the Transformer architecture, achieving a remarkable 4-bit performance that is merely 0.27 dB lower than the full-precision baseline on Set5. This extremely low performance drop confirms that SPR²Q robustly preserves high-fidelity details across diverse architectures, significantly surpassing the limitations of architecture-specific quantization methods.
>
> **Q2:Details on routing table optimization procedure.**
>
> A: We thank you for this insightful question. We will include the following details in Section 3.2.
> Optimization Method: The optimization of the static routing table is a gradient-based calibration process that employs the exact same loss function as the training stage. Specifically, we reuse the training pipeline but freeze the pre-trained backbone weights and the learned rectifier parameters, optimizing only the scalar gating weights ($\hat{g}$) to obtain the optimal weight increments for fusion.
>
> Time Cost: The process is highly time-efficient. For the MambaIRv2-light model (4-bit, ×2), calibrating on the full DF2K dataset requires approximately 40 minutes with a batch size of 8. For the larger MambaIRv2-B model, using a batch size of 1, the calibration takes around 1 hour. This calibration stage is lightweight because only the scalar gating weights are optimized, while all backbone and rectifier parameters remain frozen.
>
> Scalability: The calibration cost grows moderately with model size, as it is dominated by the model’s forward computation rather than the optimization itself. The comparison between the MambaIRv2-Light and MambaIRv2-B models shows that the increase in time is consistent with the increase in model complexity, and the overall cost remains computationally manageable (~1 hour for the larger MambaIRv2-B model).
> This indicates that the proposed calibration procedure is practical for both small and large architectures.
>
> **Q3:Quantitative verification of inference efficiency.**
>
> A:We clarify that our claim of “no additional inference cost” refers to the structural overhead of the deployed model.
> Thanks to the offline fusion mechanism, SPR²Q introduces no extra operators, no additional parameters, and no change in computational topology compared with a standard quantized model.
> 1. Theoretical Proof (Offline Fusion):
> Before inference, the learned rectifiers and routing weights are fused into the backbone weights:
>
> $$W_q = Q_{a,b}\left(W + \sum_{i=1}^N g_i \cdot \Delta W_i\right)$$
>
> Since the term $\sum g_i \cdot \Delta W_i$ is computed offline, the final deployed model keeps the exact same operator set, tensor shapes, parameter count, and FLOPs as a conventional W4A4/W2A2 quantized network.
>
> 2. Quantitative Measurements:
> Following common practice in quantization, we report FLOPs-based efficiency, which reflects the structural computation cost. SPR²Q preserves the full theoretical speed-up of quantization:
>
> |Metric|FP32|4bit|2bit|
> |-|-|-|-|
> |Model Size (MB)|3.01|1.20|1.07|
> |Compression Ratio|1.00×|2.51×|2.81×|
> |FLOPs (G)|75.6|22.0|18.2|
> |Speed-up|1.00×|3.44×|4.15×|
>
> SPR²Q can compress the model to 4 and 2 bits with the compression ratio being 2.51× and 2.81×, and speedup ratio being 3.44× and 4.15×. Crucially, these gains are achieved with zero additional inference cost, as all auxiliary parameters (rectifiers and routing weights) are fused offline.

---

> > ### Comment · Reviewer_bGTK · 2025-11-27
> >
> > Thanks authors for the detailed rebuttal. The authors' response addresses my main concerns. I keep my positive recommend. If accepted, authors must improve the final version according to the above comments.

---

### Official Review · Reviewer_jipM · 2025-10-31

**Soundness:** 3
**Presentation:** 2
**Contribution:** 2
**Rating:** 4
**Confidence:** 4

**Summary:**

his paper proposes a static priority rectifier routing quantization framework (SPR ² Q) for the low bit post training quantization (PTQ) problem of Mamba architecture image super-resolution (SR) model. This framework injects learnable compensation information through a pre quantized fusion rectifier module and combines it with a static rectifier priority routing mechanism to provide diversified compensation strategies. It effectively alleviates quantization information loss under extremely low bit (2-bit, 1-bit) settings and achieves better performance than existing SOTA methods such as PTQ4VM and Quamba on five benchmark datasets including Set5 and Urban100. Especially, it achieves PSNR improvements of 0.55dB and 1.31dB on 4-bit and 2-bit Set5 (× 2) tasks, respectively.

**Strengths:**

1. The method proposed in the paper appears to be quite general and easy to integrate into existing models.

2. The paper provides experimental results under different model settings and bits, as well as ablation experiments.

3. The paper achieved SOTA performance under different settings.

**Weaknesses:**

1. The LoRA fine-tuning in the paper requires end-to-end training, which brings additional memory and time overhead compared to other PTQ methods. And the paper uses a lightweight MambaIR model. If other versions such as MambaIR_SR are used, can Lora training be efficiently performed?

2. As the paper requires LoRA fine-tuning and weight updates, I am puzzled whether this is a traditional PTQ method and whether it is reasonable to compare it with other PTQ methods that do not require training?

3. The paper does not report the calibration resource comparison, such as GPU time and GPU memory.

4. The paper did not provide ablation experiments on LoRA's rank and router group size for calibration resources and performance.

5. Does 4bit denotes W4A4 or W4A16? And the quantization memory reduction effect and inference acceleration should be reported.

**Questions:**

See weaknesses

---

> ### Author Response · Authors · 2025-11-24
> **Response to Reviewer jipM**
>
> We thank you for the positive evaluation of our work, especially the recognition of its generality, ease of integration, and strong experimental performance. We also appreciate the constructive comments on potential limitations, which help further strengthen the paper.
>
> **Q1: Scalability of LoRA fine-tuning.**
>
> A: We have further extended our method from the smaller-scale MambaIRv2-light (774K parameters) to the larger-scale MambaIRv2-B (22.9M parameters) to validate its scalability. For the larger model, we reduced the batch size from 8 to 1 while keeping the number of iterations unchanged. In this setting, SPR²Q can still efficiently complete LoRA fine-tuning. The relevant results are presented in the table below.
> |Method|Bit|Set5(×2)||Set14(×2)||B100(×2)|| Urban100(×2)||Manga109(×2)||
> |-|-|-|-|-|-|-|-|-|-|-|-|
> |||PSNR|SSIM|PSNR|SSIM|PSNR|SSIM|PSNR|SSIM| PSNR| SSIM|
> |MambaIRv2-B|32|38.65|0.9631|34.89|0.9275|32.62|0.9053|34.49|0.9468|40.42|0.9810|
> |PTQ4VM|4|37.36|0.9569|33.07|0.9131|31.72|0.8926|31.01|0.9155|37.54|0.9734|
> |SPR²Q (Ours)|4|37.97|0.9599|33.78|0.9196|32.19|0.8999|32.69|0.9340|38.73|0.9772|
>
> On the larger MambaIRv2-B architecture, SPR²Q maintains high fidelity under 4-bit quantization, exhibiting only a marginal performance drop (0.68 dB on Set5) compared to the FP32 baseline. This confirms that our method is robust and applicable to large-scale models without prohibitive resource demands.
>
> **Q2: Reasonableness of comparison with training-free PTQ methods.**
>
> A: Quantization-Aware Training (QAT) typically requires training from scratch or extensive retraining of the full network. In contrast, SPR²Q operates directly on the pre-trained model, updating only a small set of parameters and quantization range parameters ($a, b$). This characteristic aligns it with the PTQ paradigm, as the training cost is significantly lower than that of QAT.
>
> Our design aims to address the performance degradation of PTQ in low-bit quantization. By introducing this minimal LoRA-style compensation, we have achieved significant performance improvements, largely outperforming existing PTQ methods. The comparison with PTQ methods is intended to verify that with negligible overhead, SPR²Q effectively overcomes the accuracy deterioration typical of traditional PTQ in low-bit scenarios.
>
> **Q3：Calibration resource usage (GPU time and memory).**
>
> A:Under our default experimental setting, for 4-bit quantization of the MambaIRv2-light ($\times2$) model, the rectifier group training takes approximately 4.5 hours using 4 GPUs, while the static router calibration (performed on a single GPU) takes about 40 minutes. The peak GPU memory usage is around 33 GB. Our approach is highly efficient, enabling rapid model adaptation with low time costs. We will add this information to the paper.
>
> **Q4：Ablation study on LoRA rank and router group size.**
>
> A:Our paper already includes an ablation study on the router group size. In addition, we have conducted further experiments on different LoRA ranks r=2,4,16, the corresponding results are provided below.
>
> |r|Trainable Params|Set5(×2)||Urban100(×2)||
> |-|-|-|-|-|-|
> |||PSNR|SSIM|PSNR|SSIM|
> |2|155K|37.63|0.9585|31.36|0.9208|
> |4|309K|37.67|0.9588|31.48|0.9221|
> |8|616K|37.72|0.9589|31.53|0.9223|
> |16|1231K|37.74|0.9591|31.70|0.9240|
>
> Performance improves as the rank increases, but the benefits largely plateau beyond r=8.
> During these experiments, GPU memory usage remained stable at ~33 GB across all ranks, because memory in SR models is dominated by high-resolution activations rather than parameter size. The lightweight nature of SPR²Q comes from freezing the backbone, so no gradients or optimizer states are maintained for the main network. As a result, even with increasing rectifier size, the actual computation and memory remain far lower than full-model fine-tuning. We also note that on the larger MambaIRv2-B model, only 5.1M rectifier parameters are trained compared with the 23M frozen backbone, further confirming that SPR²Q remains lightweight even on high-capacity architectures.
>
> **Q5:Clarification of “4-bit” definition and reporting memory/latency benefits.**
>
> A:In our method, 4-bit refers to W4A4 quantization. The memory reduction and theoretical inference acceleration brought by our quantization are presented in the following table. All parameter statistics are obtained on MambaIRv2-light, and the FLOPs are measured using an input resolution of (180, 320) under the 4× SR setting.
>
> |Metric|FP32|4bit|2bit|
> |-|-|-|-|
> |Model Size (MB)|3.01|1.20|1.07|
> |Compression Ratio|1.00×|2.51×|2.81×|
> |FLOPs (G)|75.6|22.0|18.2|
> |Speed-up|1.00×|3.44×|4.15×|
>
> SPR²Q can compress the model to 4 and 2 bits with the compression ratio being 2.51× and 2.81×, and speedup ratio being 3.44× and 4.15×. Benefiting from the Offline Static Routing mechanism, SPR²Q achieves the theoretical upper limit of compression and speedup while maintaining superior performance.

---

### Author Response · Authors · 2025-12-02
**Summary comment for the Area Chair**

Dear Area Chair,

We thank all reviewers and the area chair for their thorough evaluation and constructive feedback. These comments have been highly valuable for improving the rigor and completeness of our work. We have revised the manuscript and added supplementary experiments based on the reviewers' feedback. To help the AC quickly understand the key updates made during the rebuttal process, we provide a concise summary below.

First, our method, SPR²Q, demonstrates strong advantages in low-bit super-resolution quantization. Reviewers generally recognized its broad applicability, ease of integration, and strong empirical performance. Specifically, SPR²Q maintains high reconstruction fidelity under both 4-bit and 2-bit settings, outperforming existing approaches. Owing to the proposed Offline Static Routing mechanism, all auxiliary parameters are merged before inference, introducing no additional operators on the deployment side and enabling zero inference overhead. Moreover, the method exhibits high architectural generality, supporting both Mamba-based and Transformer-based super-resolution models.

During the rebuttal, we systematically addressed concerns regarding generalization, comparisons with related quantization baselines, and the underlying mechanism. We added experiments on larger models and Transformer architectures, further validating the stability and effectiveness of SPR²Q across different network designs. Additionally, we included comparisons on Transformer-based SR models against three representative SOTA quantization methods, showing that SPR²Q consistently achieves superior performance. We also reported resource usage, acceleration and compression benefits, and provided a frequency-domain analysis explaining the rectifiers’ specialization in texture and structural information. Expanded ablation studies further resolved the majority of technical questions. An anonymized code repository has also been released to ensure reproducibility: https://github.com/momo5-a11/SPR2Q.

Regarding reviewer feedback, most reviewers confirmed that their major concerns were resolved after reading the rebuttal. Several reviewers maintained their positive recommendations, and one reviewer raised their score. In response to the request for SwinIR quantization baseline comparisons, we also provided additional experiments under unified configurations.

We believe that these additional experiments, theoretical clarifications, and the release of our code have significantly enhanced the completeness and persuasiveness of the paper, addressing the vast majority of the reviewers' questions. We appreciate the AC's time and consideration.

Best regards,

Authors

---

### Meta-Review · Area_Chair_hN7i · 2026-01-04

**Summary:**

The major concerns from the reviewers are about the evaluation and the details of the method. Three reviewers' (bGTK, xpnk, xceu) concerns have been addressed while one reviewer (jipM) did not reply. However, the rebuttal has addressed the reviewer's concern. Therefore, my suggested decision is accept (poster).

**Reviewer Concerns:**

Reviewer jipM's concerns about scalability, reasonableness, calibration, and ablation have been replied and I think the concerns are addressed.

Reviewer bGTK's concerns about evaluation, optimization, and overhead have been addressed by the rebuttal.

Reviewer xpnk's concerns about evaluation, analysis, experiments, and writing have been addressed by the rebuttal.

Reviewer xceu's concerns about evaluation, complexity, and rank have been addressed by the rebuttal.

**Reviewer Scores:**

Reviewer jipM may increase the score to 6 as the concerns are all addressed.

Reviewer bGTK and Reviewer xpnk would keep the score as 6.

Reviewer xceu would increase the score to 6 as the concerns are all addressed.

---

### Decision · Program_Chairs · 2026-01-26

Accept (Poster)